# Psychosocial Factors Predicting Resilience in Family Caregivers of Children with Cancer: A Cross-Sectional Study

**DOI:** 10.3390/ijerph18020748

**Published:** 2021-01-17

**Authors:** Filiberto Toledano-Toledano, David Luna, José Moral de la Rubia, Silvia Martínez Valverde, Carlos Alberto Bermúdez Morón, Marcela Salazar García, Mario José Vasquez Pauca

**Affiliations:** 1Evidence-Based Medicine Research Unit, Hospital Infantil de México Federico Gómez, National Institute of Health, Dr. Márquez 162, Doctores, Cuauhtémoc, Mexico City 06720, Mexico; cbermudez494@gmail.com (C.A.B.M.); marcelasalazargarcia@hotmail.com (M.S.G.); 2Comisión Nacional de Arbitraje Médico. Mitla No. 250-10 Piso, esq. Eje 5 Sur (Eugenia), Col. Narvarte, Benito Juárez, Mexico City 03020, Mexico; xeurop@hotmail.com; 3Facultad de Psicología, Universidad Autónoma de Nuevo León, Dr. Carlos Canseco, 110, Esq. Dr. Aguirre Pequeño, Col. Mitras Centro, Monterrey 64460, Mexico; jose_moral@hotmail.com; 4Centro de Estudios Económicos y Sociales en Salud, Hospital Infantil de México Federico, Gómez, National Institute of Health, Dr. Márquez 162, Doctores, Cuauhtémoc, Mexico City 06720, Mexico; simava2001@yahoo.com.mx; 5Facultad de Trabajo Social, Universidad Nacional del Altiplano Puno, Floral 1153, Puno 21001, Peru; mvasquezpauca@gmail.com

**Keywords:** resilience, family caregivers, children, cancer, depression, México, caregiver burden, stress, well-being, quality of life

## Abstract

Chronic diseases in childhood can affect the physical and mental health of patients and their families. The objective of this study was to identify the sociodemographic and psychosocial factors that predict resilience in family caregivers of children with cancer and to define whether there are differences in the levels of resilience derived from these sociodemographic variables. Three hundred and thirty family caregivers of children with cancer, with an average age of 32.6 years were interviewed. The caregivers responded to a battery of tests that included a questionnaire of sociodemographic variables, the Measuring Scale of Resilience, the Beck Depression Inventory, the Inventory of Quality of Life, the Beck Anxiety Inventory, an interview of caregiver burden and the World Health Organization Well-Being Index. The main findings indicate that family caregivers of children with cancer reported high levels of resilience, which were associated positively with quality of life, psychological well-being and years of study and associated negatively with depression, anxiety and caregiver burden. The variables that predicted resilience in families of children with cancer were quality of life, psychological well-being, depression and number of children. Family caregivers who were married and Catholic showed higher resilience scores. We conclude that being a caregiver in a family with children with cancer is associated with symptoms of anxiety and with depressive episodes. These issues can be overcome through family strength, well-being, quality of life and positive adaptation processes and mobilization of family resources.

## 1. Introduction

Globally, cancer is the second-highest cause of death in children between the ages of 5–14 years, following accidents. Furthermore, there has been an increase in childhood cancer in recent decades [1]. Acute lymphoblastic leukemia (26%), brain and central nervous system tumors (21%), neuroblastoma (7%) and non-Hodgkin lymphoma (6%) are among the most common types of cancer in children. The five-year survival of children diagnosed with cancer is greater than 80%, and the prevalence of cancer in children is greater in White and Hispanic ethnic groups [2] based on mortality data from the Mexican National Institute of Statistics (INEGI by its acronym in Spanish) [3].

In Mexico, childhood cancer is a public health problem and is the leading cause of death in children aged 5–14 years. The most common types are acute lymphoblastic leukemia, acute myeloblastic leukemia, non-Hodgkin lymphoma and Hodgkin disease. Annually, 5000 cases of childhood cancer are diagnosed, representing 5% of the total number of cancer cases diagnosed, mostly in advanced stages of the disease. The five-year survival of children diagnosed with cancer in Mexico is 56% [4].

Childhood cancer has serious repercussions on the physical and psychological health of pediatric patients, their families and their caregivers [5,6]. Caregiving can be experienced as a stressful process, which can cause psychological and physical consequences [7]. The effects and consequences experienced by families caring for children with cancer include the risk of developing anxiety, depression and parental stress [8,9,10]; caregivers of such children experience a host of stressful events, the most common of which are poor health [11], caregiver burden and burnout [12,13]; childhood cancer patients and their families frequently experience psychosocial distress associated with cancer and its treatment [14], decreases in psychological well-being [15,16], and effects on their quality of life [17,18].

There is also a set of contextual factors and sociodemographic characteristics in family caregivers of pediatric patients that increase the risk of suffering repercussions on physical and psychological health [19]. The main demographic variables include gender [20], unemployment [8,9], low income [21], low education levels [22], the social support networks [23], caregiver marital status [24], number of children in the family [25], child age [26] and psychosocial profile of family caregivers [27]. Context factors comprise time elapsed since diagnosis [28], development of the chronic disease [8,29,30,31], type of cancer [32] and duration and impact of care [33,34].

Among the contextual factors, two of them stand out. Religious support is one of the components of resilience with the greatest weight in chronically and terminally ill patients [35]. Furthermore, among family caregivers of children with chronic diseases, it is highly relevant. Religious beliefs and the practice of prayer provide hope and confidence. They even open a path to find meaning in the adverse situation that is experienced [36]. On the other hand, the married marital status is another source of support both in material and emotional aspects [24].

Researchers and clinicians have been pointing out that family caregivers of children with cancer need to develop the ability to adapt and operate optimally to overcome adversity [36,37]. This ability has been explained through the construct of resilience [16,37]; resilience is a complex construct that can change over time, but it can also facilitate the development of interventions aimed at promoting resilience in the families of children with cancer [38], which in the context of disease has been defined as the ability of individuals to maintain or regain their physical and emotional health in the face of risk situations that entail adversity [39]. Thus, resilience is an adjustment and adaptation process to the demands of chronic illness by the family [40]. Family resilience, that is, the potential resources of the family system, has been considered due to numerous individual studies that highlight the crucial influence of relationships with significant other people in mediating adaptation and recovery [41].

Family resilience in the face of disease is a process of positive adaptation despite the loss of health. This process involves the development of vitality and skills to overcome the negative effects of adversity, risk, and vulnerability caused by disease [42]. Resilience helps individuals overcome the diagnosis crisis, lessens the impact of the tasks entailed by medical care and treatment and improves the psychosocial adjustment and well-being of caregivers during the child’s chronic illness [9,28,38,43].

Although there is evidence that highlights the relevance and usefulness of conducting research on resilience in families of children with cancer [6,38,40,44,45,46], relatively few studies have focused on identifying the psychosocial and sociodemographic factors that predict resilience in family caregivers of children with cancer. Consequently, obtaining useful information to generate interventions with theoretical and empirical foundations aimed at developing family resilience to the chronic disease of their children. Parents of children with cancer are at risk for negative psychosocial outcomes, and those with low resilience may be at higher risk. Interventions aimed at promoting resilience resources can provide a novel and complementary approach to improve outcomes for families dealing with pediatric cancer [46]. Therefore, to address the theoretical issues and empirical evidence available, the first objective of this study was to identify the psychosocial factors that predict resilience in family caregivers of children with cancer. The second objective was to define whether there are differences in resilience levels based on sociodemographic variables.

## 2. Materials and Methods

### 2.1. Participants

A total of 330 family caregivers of hospitalized children with cancer were interviewed at the Hospital Infantil de México Federico Gómez, National Institute of Health, in Mexico City. A non-experimental, transversal, *ex post facto* study was conducted, using a convenience and non-probabilistic sampling technique. The sample included women (82%) and men (18%) aged between 18–63 years, with an average age of 32.60 and standard deviation (SD) of 8.59. The inclusion criteria for the study were (1) being a family caregiver of a child who was receiving cancer treatment, (2) being at least 18 years of age and (3) having signed an informed consent. The exclusion criteria were (1) inability to read and write and (2) refusal to participate in the study. The deletion criteria included partial or incomplete responses to the psychosocial measurement instruments. The pediatric patients included both girls (52%) and boys (48%), aged between 1–17 years, with an average age of 6.33 and standard deviation of 5.13. In most cases, the time elapsed since cancer diagnosis ranged from one week to one year (68.5%), and the hospitalization period was one week to one month (83.9%).

### 2.2. Instruments

A battery of test instruments, including a questionnaire with sociodemographic variables to conduct research on families of children with chronic diseases and six self-report instruments measuring psychosocial variables (resilience, depression, anxiety, quality of life, caregiver burden and psychological well-being), were used. To guarantee the accuracy of the data obtained, the instruments were validated in the Mexican population and with families of children with chronic diseases.

*Sociodemographic variables questionnaire (Q-SV) for research in family caregivers of children with chronic diseases* [19]. This questionnaire contains 20 items that evaluate information on sociodemographic, medical, sociocultural and family variables in families of children with chronic diseases. For this study, the diagnosis, age and sex of the patient and caregiver, the relationship between the two (mother, father or another family member), the educational level (no schooling, primary education, secondary education, undergraduate education, postgraduate education), occupation (housemaker, worker, trader, employee, student, pensioner, unemployed), marital status (married, living together, separated, divorced, single parent, widowed), years of partnership, number of children, type of family (nuclear, semi-nuclear, extended, single-parent), family life cycle (with young children, with school-age children, with adult children), social support networks (family, friends, religion, institutions, government), religion (Catholic, Christian, none) and monthly income were determined.

*Measurement Scale of Resilience in Mexicans (RESI-M;* [47]), validated in family caregivers of children with cancer [42]. This scale contains 43 four-point Likert-type items, ranging from 1 “strongly disagree” to 4 “strongly agree,” and measures the level of overall resilience and five dimensions: Strength and self-confidence (19 items), Social competence (eight items), Family support (six items), Social support (five items) and Structure (five items). Among the 330 family caregivers in the present study, the overall internal consistency of the 43 items of the RESI-M was excellent through Cronbach alpha coefficient (α = 0.95). Table 1 shows the internal consistency of its five factors.

Beck Depression Inventory II (DBI-II; [48]), validated in a population of family caregivers of children with chronic diseases [49]. This inventory includes 21 items, each with four statements that assess depressive symptomatology and episodes. It uses a rating scale from 0–3, where the higher the score, the higher the level of depression. The level of depression is interpreted as follows: minimum from 1–4, mild from 5–13, moderate from 14–27 and severe from 28–63 points. Among the 330 family caregivers in the present study, the overall internal consistency of the 21 items was excellent through Cronbach alpha coefficient (α = 0.91).

Beck Anxiety Inventory (BAI; [50]), this instrument has been validated in family caregivers of children with cancer, by Toledano-Toledano et al. [51]. Through 16 items, this inventory assesses anxious symptomatology using a four-point scale, ranging from 0 “Little or nothing” to 3 “Severely.” The level of anxiety obtained is minimum (1–5 points), mild (6–15), moderate (16–30) and severe (31–63). In the present sample, the overall internal consistency of the 21 items was excellent (α = 0.92).

Inventory of Quality of Life WHOQOL-BREF [52], validated in a Mexican population [53]. This inventory includes 26 five-point Likert-type items ranging from 1–5. Two items constitute general questions about quality of life, and the remaining are grouped in the following dimensions: Physical health (seven items), Psychological health (six items), Social relations (three items) and Environment (eight items). Among the 330 family caregivers in the present study, the overall internal consistency of the 26 items was excellent (α = 0.92). Table 1 shows the internal consistency of its three factors.

Zarit Burden Interview (ZBI [54]), validated in a Mexican population [55]. This tool assesses the subjective burden, attitudes and emotional reactions of the caregiver when faced with the responsibility of care and the perception of the situation. It contains 22 items distributed across three factors: Impact of caregiver (13 items), Interpersonal relationship (six items) and Self-efficacy expectations (three items). The scores of the items range from 0 “Never” to 4 “Always.” In the present study, only the ZBI total score was used, and its overall internal consistency was excellent among the 330 family caregivers (α = 0.90).

World Health Organization Well-Being Index (WHO-WBI; [56]). This instrument was validated specifically for this study, with the sample described above, using the Forward-Backward Translation strategy. It contains nine four-point Likert-type items, ranging from 0 “Never” to 3 “All the time”. It consists of two factors that explain 62.53% of the total variance, the first with five items related to psychological well-being (α = 0.83) and the second with four items related to physical well-being (α = 0.81). The overall internal consistency of its 9 items was good (α = 0.89). Table 1 shows the internal consistencies of the two factors.

### 2.3. Ethical Considerations

This study is a part of the research project HIM/2015/017/SSA.1207 “Effects of mindfulness training on psychological distress and quality of life of the family caregiver,” which was approved on 16 December 2014, by the Research, Ethics, and Biosafety Commissions of the Hospital Infantil de México Federico Gómez, National Institute of Health, in Mexico City. While conducting this study, the ethical rules and considerations for research with humans currently enforced in Mexico [57] and those outlined by the American Psychological Association [58] were followed. All family caregivers were informed of the objectives and scope of the research and their rights according to the Declaration of Helsinki [59]. The caregivers who agreed to participate in the study signed an informed consent letter. Participation in this study was voluntary and did not involve payment.

### 2.4. Procedure

The family caregivers were interviewed by the primary author of this study in the wards of the Hematology-Oncology Service of the Hospital Infantil de México Federico Gómez, National Institute of Health. All family caregivers who were asked to participate agreed to signed informed consent forms and answer to the instruments. None refused. Participants did not face any consequences for withdrawing their consent, as specified on the informed consent sheet. They answered the instruments individually during a single session. Before collecting the completed instruments, the interviewer checked that there were no questions without answers. If there were questions without answers, the participant was asked to respond to them, and in this way, we managed to avoid the presence of missing values.

### 2.5. Data Analysis

The data were analyzed with SPSS v. 24 for Windows (IBM, Armonk, NY, USA). A descriptive analysis of the sociodemographic characteristics and psychosocial aspects of the population, which included frequency of cases (%) for non-metric variables and averages (*M*) and standard deviation (*SD*) for metric variables, was performed. Given its non-normal distribution (Shapiro–Wilk test, *p* < 0.05), in order to identify the associations between the score obtained while measuring resilience and the psychosocial variables, as well as with the sociodemographic variables years of partnership, years of study (i.e., schooling converted to years: no schooling: 0, primary education: 6, secondary education: 9, high school: 12, undergraduate degree: 16, postgraduate degree: 18), number of children and age of the pediatric patient, correlation analyses were conducted using Spearman’s rho. To analyze the correlation between categorical data and level of resilience, total resilience and resilience by factor, as presented by the RESI-M, were divided into tertiles based on the categories mild, moderate and high. Then, chi-square tests calculating Kramer’s V coefficient and Pearson’s residuals were conducted in those associations that proved to be significant.

A predictive model of resilience was calculated using stepwise linear regression. The predictor variables included the sociodemographic variables age, years of partnership, years of study, number of children and age of the pediatric patient. The scores obtained in the scales DBI-II, BAI, WHOQOL-BREF and WHO-WBI were considered among the psychosocial variables. The explanatory strength of this model was determined by *R*^2^ adjusted, improvement of fit was tested by the *F*-test, and the statistical tolerance (TOL) and variance inflation factor (VIF) were checked to control for collinearity. Residual analysis included the Shapiro–Wilk test to check for its normality, the Durbin–Watson test to check for autocorrelation of consecutive residuals and Cook’s distance to identify influential values.

Following Cohen’s cut-off points, a standardized regression weight (*β*) or a correlation (*rho*) below 0.10 was interpreted as a trivial association, values ranging from 0.10–0.299 were low, those from 0.30–0.499 were medium, those from 0.50–0.699 were high, those from 0.70–0.899 were very high, and values equal to or greater than 0.90 were unitary [60,61]. Differences in the overall score and in each factor’s score assessed through RESI-M were analyzed with a two-tailed Mann–Whitney U test and the coefficient of probability of superiority (PS_est_) as an index of effect size for the sociodemographic variables sex of the caregiver, sex of the pediatric patient and marital status. The Kruskal–Wallis test and Nemenyi post hoc test were used to analyze the Religion variable. For all tests, significant results were considered at *p* value ≤ 0.05.

## 3. Results

### 3.1. Characteristics of the Family Caregivers

The results indicated that the highest percentage of caregivers was represented by mothers (77.3%) or fathers (16.1%) of pediatric patients with cancer diagnosis (43.6%), leukemia (17%), tumors (22.1%), osteosarcoma (8.2%) or neuroblastoma (9.1%). In most cases, secondary education was the maximum level of study achieved by caregivers (47.6%), and they engaged in household work (67%). They generally came from nuclear families (50.9%) who practiced the Catholic religion (81.8%) and whose main support network was their own family (82.1%). A high percentage of family caregivers were married or in domestic partnerships (76.6%), with an average of 9.61 years (SD = 7.38) as a couple and 2.39 (SD = 2.15) children, mostly of school age or adolescents (61.5%). Among participants, 82.4% of them reported a maximum monthly income of US$137.10.

Table 2 shows the levels of depression and anxiety reported by the sample of caregivers, and Table 1 presents the average values obtained per factor for each of the psychosocial measurement instruments, the range of scores for the factors in each scale and the values of internal consistency estimated by Cronbach’s alpha. In general, family caregivers showed minimum to mild levels of anxious symptomatology and mild to moderate depression scores, as well as elevated scores in the other psychosocial variables.

### 3.2. Association between Resilience, Psychosocial Variables and Sociodemographic Characteristics

Positive associations were found between the scores of the variable resilience and the psychosocial variables quality of life (*rho* = 0.51), well-being index (*rho* = 0.51) and years of study (*rho* = 0.14), and negative correlations appeared with the variables depression (*rho* = −0.46), anxiety (*rho* = −0.29) and caregiver burden (*rho* = −0.33), all with a *p* < 0.01. Table 3 presents the correlation between these variables and the RESI-M factors. It shows that significant correlations obtained values that indicate a low association (*rho =* |0.11–0.27|) in a higher number of cases. In addition, this level of association appeared in the variables assessed and in the social support and structure factors of the RESI-M. Correlations indicating a medium association (*rho =* |0.31–0.46|) were found in the three remaining factors, and a high association (*rho =* 0.51) occurred only between the factor of strength and self-confidence and the variables related to quality of life and psychological well-being.

Chi-square tests revealed an association between the patient’s sex and the strength and self-confidence factor (χ^2^ = 6, df = 2, V = 0.13, *p* = 0.05), indicating a greater number of resilient family caregivers of male patients than female patients. For the structure factor, a lower number of Christians with moderate resilience was detected and a greater number with mild resilience (χ^2^ = 9.5, df = 4, V = 0.12, *p* = 0.05). Finally, the social support factor had a greater number of caregivers with moderate resilience when their incomes were low, but with a high level of resilience with higher incomes (χ^2^ = 9.08, df = 2, V = 0.01, *p* < 0.01). It should be noted that the V coefficient refers to a weak association in all cases. For the remaining variables, no significant associations were detected either in relation to the total level of resilience or to the level of individual factors (*p* > 0.05).

### 3.3. Multiple Linear Regression for Predicting Resilience

Although a first model was generated, the residual analysis did not satisfy the normality assumption (Shapiro–Wilk, *p* < 0.01). Because of this situation, the response variable data were transformed to natural logarithm units (i.e., Ln(Resilience)), and the model was then calculated again. Out of the five psychosocial variables (depression, anxiety, quality of life, caregiver burden and well-being) and the only sociodemographic variable (years of study) that correlated with the total level of resilience, all except one (anxiety) were kept in a regression model that accounted for 41% of the variance in caregiver resilience (*R*^2^ = 0.419; adjusted *R*^2^ = 0.410). The results from the ANOVA table showed that the full model was statistically significant (*F* (5, 324) = 46.70, *p* < 0.001). In addition, there is no evidence for collinearity among predictors (TOL = 0.595–0.943 and VIF = 1.060–1.680). The regression weights, with effect sizes ranging from trivial to medium, and collinearity statistics are presented in Table 4. The Durbin–Watson test (with data ordered in its collection sequence) revealed that the residuals were independent (*DW* = 2.09), and the Shapiro–Wilk test showed its normal distribution (*p* > 0.05). These results indicate that the assumptions of independence and homoscedasticity were not violated. Finally, Cook’s distance also did not reveal any problematic data since the values found were all below the cut-off of 1.0 (*d* = 0.000–0.111). The statistical power of the analysis performed was estimated using a post hoc test, taking into account the size of the sample used (*N* = 330), the probability of committing a type I error (α = 0.05) and a medium effect size (*ES* = 0.15) with five predictors, resulting in a statistical power (1 − β) = 0.99 [59]. The set of analyses described above provide evidence about the validity of the model generated.

### 3.4. Differences in the Level of Resilience

No differences were detected in the level of resilience when assessing overall scores of men and women, sex of the pediatric patient or caregiver religion (*p* > 0.05). Differences appeared when married family caregivers and those who share a domestic partnership were compared, the latter being less resilient with respect to the former (*U* = 6677.50, *Z* = −2.160, *p* < 0.05). A comparative analysis among the factors in the resilience scale and the sex of the caregiver and the pediatric patient (*p* > 0.05) did not report significant differences. There were differences in the social competence factor, in which married caregivers scored higher than caregivers in domestic partnerships (*U* = 6321, *Z* = −2.806, *p* < 0.05). The four remaining factors presented an almost significant tendency (*p* = 0.07), which showed higher scores for married family caregivers. Finally, differences were found in the scores obtained in the structure factor in relation to religion of the caregiver (χ^2^ = 6,790, *p* < 0.05, *N* = 329). Post hoc multiple comparisons showed differences between Christians and Catholics, the latter being those who scored higher in this factor (*p* < 0.05).

## 4. Discussion

The objective of this study was to identify the sociodemographic and psychosocial factors that predict resilience in family caregivers of children with cancer. A multiple regression model revealed that the psychosocial factors quality of life and psychological well-being and the sociodemographic variable years of study were positive predictors of resilience and that depression and caregiver burden represented negative predictors. The data also revealed that Catholic family caregivers presented greater resilience, specifically in the structure factor, i.e., the ability of family caregivers to organize, to plan activities and time and to have rules and activities even in times of adversity. Married caregivers had the highest values in the social competence factor, which points to their ability and competence to relate to others, to make new friends easily, to make people laugh and to enjoy a conversation. These conclusions refer to the second objective, which entailed detecting differences in resilience based on sociodemographic variables.

Some characteristics of the profile of family caregivers of children with cancer include being a woman, typically the mother of the patient, which is consistent with the results obtained in other investigations [12,62,63]; who is in a productive age (*M* = 32.60 years), also consistent with data from other studies [64,65], who is dedicated to household work and has no formal income.

Considering the findings of this study on the characterization of caregivers of children with cancer, it is possible to assume that there is limited involvement of the father as a caregiver, most likely because he would be working to gather consumables and cover the out-of-pocket expenses entailed by chronic diseases. Most family caregivers are married or living in domestic partnerships, characterized by constituting nuclear families, with low income. This characteristic may be because the highest level of studies achieved is secondary education, and only one member of the family receives financial remuneration for his or her work. This low economic status was associated with moderate levels of resilience. Conversely, families who reported high levels of resilience had greater economic incomes. These findings correspond to the structure factor measured by the RESI-M. In addition, as reported in previous studies [66], the family itself constitutes the support network in most cases.

In this group of family caregivers, the variables support networks, level of parental education and income are fundamental to decreasing the impact and effects of care. The same happens with the adaptation to the child’s illness. These findings are consistent with those reported in previous research, which suggests that family support, economic income and educational level of the caregiver are associated with family caregiver morbidity [21,22,23,25]. Similarly, recent research has reported benefits in quality of life [5] and the decline in psychosocial risks when the caregiver of the child perceives greater family support [6,44].

The analysis of anxiety and depression levels indicated that they remain within minimal to moderate levels. The low percentage of family caregivers with high levels of both of these symptoms, approximately 12%, is similar to that reported in other studies with families of children with chronic diseases [12,38]. The findings from this study suggest that such results may be due to the time elapsed since the child was diagnosed with cancer, given that it is relatively short (one year maximum for 68.5% of the sample), as well as the length of the hospitalization period (from one week to one month for 83.9% of the sample). Therefore, the results of this study demonstrated that, despite the adversity caused by the disease and the vulnerability to which they are exposed, family caregivers of children with cancer reported high levels of resilience and other protective factors that contribute to their psychosocial adjustment. These findings are consistent with previous data pointing to the ability of the families of children with cancer to adapt and overcome the impact of chronic illness [40,45,67].

In addition, the findings of this research indicated that the sociodemographic variable years of study showed a positive correlation with resilience. However, the degree of association between these variables was low. These results suggest a low participation of one of the variables with respect to the level of the other. More importantly, the correlations with medium to high associations focused on the strength and self-confidence, social competence and family support factors in the resilience scale. According to the regression model, these findings confirm that the psychosocial variables that predict family resilience in the face of illness are quality of life, psychological well-being and depression and that the sociodemographic variable that predicts the levels of resilience in family caregivers of children with cancer is years of study. These findings may result from the greater contribution of the psychosocial variables to the specific factors of strength and self-confidence, social competence and family support and minimally to the remaining factors (i.e., social support and structure).

Finally, the findings of this study confirm that there are no differences between men and women with regard to the levels of resilience perceived by family caregivers of children with cancer. These results are consistent with previous research [6,47,68], which reported no differences in sex when measuring and assessing resilience. It was also found that caregivers who professed the Catholic religion scored highest in the structure factor, and the same happened with married caregivers in relation to the social competence factor. These findings are consistent with those reported in recent studies that have evaluated participation in religious activities and/or spirituality, as well as marital status and resilience [69,70]. Despite these findings, other studies have suggested the need for further research on mental health, spirituality and the processes of positive adaptation [36,71].

Among the limitations of this study is the cross-sectional nature of the design, which prevents establishing causal relationships or verifying the consistency of results over time. To do so, it would be necessary to conduct explanatory research studies with longitudinal design. Second, it is likely that the indicators used as effects of care—for example, anxiety—do not reflect the consequences in the psychological health of the caregiver, and it would be important to include psychological stress instead. Third, the present study was based on a single informant and did not have multiple informants among relatives, as is usual in a survey study. It could be considered that this fact may affect the reliability of the data. However, the measurement instruments have their reliability estimated through internal consistency in the current sample and other samples and these values are good or very good. They have even shown stability through test-retest reliability. Finally, it is suggested for future studies to include other variables that have been reported previously as important predictors of resilience in contexts of pediatric chronic illness, such as family functioning, family support and social support networks.

## 5. Conclusions

The main implication of this study is that, having identified the sociodemographic factors that predict resilience, clinical programs and psychological intervention with families of children with chronic illnesses should include sociodemographic variables and variables related to the sociocultural context of families, with the aim of contributing to family strengths, resources and processes of positive adaptation in family caregivers of children with cancer. In this regard, we suggest performing interventions aimed at strengthening the skills required for promoting social relations and the confidence required to achieve personal goals and face challenges. Creating programs focused on promoting the physical and emotional health of patients and their families in this context is also recommended in order to reduce psychosocial risks and dysfunction and to develop healthy lifestyles conducive to quality of life. These programs could be coordinated by federal and state health secretaries and administered by hospitals and tertiary care centers.

The present study has emphasized personal resilience in coping with the care of a chronically ill child. This approach gives more weight to the family and parents than the social health system. However, this last aspect should not be overlooked. It is important that public health policies guarantee the right to health of citizens and promote social resilience to face the care of children with chronic diseases. We suggest that future studies adopt the social resilience approach, which would be novel in this field of study [72]. Likewise, other relevant questions that should be addressed in future research are how the health system, and health professionals, can contribute to helping these families with children with chronic diseases. Is it necessary to deepen the studies to see how families are resilient with good health systems and health professionals attentive to this aspect?

## Figures and Tables

**Table 1 ijerph-18-00748-t001:** Average scores obtained from the factors of each measurement instrument.

Scales and Their Factors	Items	Range	M (SD)	α
Resilience Scale for Mexicans	43	-	-	0.95
Strength and self-confidence	19	19–76	59.85 (8.23)	0.93
Social competence	8	8–32	22.77 (4.1)	0.87
Family support	6	6–24	19.72 (3.2)	0.87
Social support	5	5–20	16.03 (3.2)	0.90
Structure	5	5–20	14.32 (2.54)	0.76
Quality of Life Inventory	26	-	-	0.90
Physical health	7	7–35	20.85 (3.24)	0.71
Psychological health	6	6–30	20.72 (3.54)	0.69
Social relations	3	3–15	7.06 (1.69)	0.67
Environment	8	8–40	24.84 (4.42)	0.75
Zarit Burden Interview	22	-	-	0.90
Impact of caregiver	13	0–52	12.62 (8.24)	0.89
Interpersonal relationship	6	0–24	2.61 (3.23)	0.84
Self-efficacy expectation	3	0–12	7.44 (3.18)	0.79
Scale of Psychological Well-Being	9	-	-	0.89
Factor 1 Personal well-being	5	0–12	6.26 (1.79)	0.83
Factor 2 Subjective well-being	4	0–15	12.52 (3.58)	0.81

*Note*: M = arithmetic mean, SD = standard deviation, α = Cronbach alpha coefficient.

**Table 2 ijerph-18-00748-t002:** Levels of anxiety and depression in caregivers (*N* = 330).

Level	Anxiety		Depression	
	%	M (SD)	%	M (SD)
Minimum	28.2	3.06 (1.38)	16.6	2.36 (1.17)
Mild	37	9.50 (2.75)	37.6	8.83 (2.61)
Moderate	22.7	22.22 (4.61)	34.2	18.92 (3.94)
Severe	12.1	39.57 (7.22)	11.5	33.50 (5.42)

*Note*: M = arithmetic mean, SD = standard deviation, % = percentage.

**Table 3 ijerph-18-00748-t003:** Correlations with resilience factors.

Psychosocial and Sociodemographic Variables	Resilience Factors				
	Strength and Self-Confidence	Social Competence	Family Support	Social Support	Structure
Quality of life	0.51 **	0.34 **	0.37 **	0.32 **	0.18 **
Psychological well-being	0.51 **	0.33 **	0.33 **	0.25 **	0.27 **
Years of study	0.08 ^ns^	0.14 **	0.11 *	0.18 **	0.02 ^ns^
Depression	−0.46 **	−0.34 **	−0.38 **	−0.17 **	−0.19 **
Anxiety	−0.27 **	−0.27 **	−0.22 **	0.09 ^ns^	−0.15 **
Caregiver burden	−0.31 **	−0.24 **	−0.22 **	−0.23 **	−0.12 *

*Note*: Probability values in a two-tailed test * *p* < 0.05; ** *p* < 0.01; ^ns^ (not significant) *p* ≥ 0.05.

**Table 4 ijerph-18-00748-t004:** Regression model predicting resilience in family caregivers, Sample size (*N*) = 330.

Predictor Variables	*B* (95% *CI*)	β	*t*	*p*	TOL	VIF
Quality of life	0.003 (0.002, 0.004)	0.291	5.54	0.0001	0.652	1.533
Psychological well-being	0.007 (0.004, 0.009)	0.242	4.77	0.0001	0.698	1.433
Depression	−0.002 (−0.004, −0.001)	−0.189	−3.43	0.001	0.595	1.680
Years of schooling	0.004 (0.0002, 0.007)	0.093	2.12	0.034	0.943	1.060
Caregiver burden	−0.001 (−0.002, 0.000007)	−0.096	−1.98	0.048	0.766	1.306

*Note: B* = unstandardized coefficients, *CI* = confidence interval, β = standardized coefficients, *t* = Student’s *t*-test statistic; *p* = *p*-value for a two-tailed *t*-test, TOL = statistical tolerance, and VIF = variance inflation factor. Method: Stepwise.

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
