# Peer review of "Psychosocial Factors Predicting Resilience in Family Caregivers of Children with Cancer: A Cross-Sectional Study"

_ijerph, 2021, doi:10.3390/ijerph18020748_

Round 1
Reviewer 1 Report
Thank-you for the opportunity to review your manuscript on resilience in carers of children with cancer.
Although the research has been thoughtfully and rigorously conducted, and the manuscript is generally well written and clearly presented, I have some concerns about its overall value.
It does not seem to add anything beyond what we already know - i.e. that carers need more support. It also doesn't engage with literature that is critical of the growing focus on resilience - that literature suggest that an emphasis on personal/familial resilience only puts further pressure on carers when what's really required is resilient systems/societies.
I'd encourage you to give some serious and critical thought to whether or not this research makes a meaningful contribution to current academic debate or offers anything new to inform clinical & social practice.
Author Response
please see the attacment

Reviewer 2 Report
As stated by the authors, resilience is a very complex construct. One of the major debates on resilience research concerns its conceptualization as a personality trait or as a process. According to the literature review of this manuscript, it is consider as a result in terms of adjustment as a response to the adversity of the cancer diagnosis of the child; furthermore, it is described not as individual, but as familiar. Although, the study is cross sectional, and the answers to the scale are given only by the caregiver, mostly the mother. How do authors consider that these limitations may have an impact on the results observed: not having a baseline of resilience levels and even other outcomes, before the diagnosis, and also not having multiple informants among relatives but to rely on a single informant.
Furthermore, it is possible that some self-selection of the caregivers may have occurred, that may influenced sample composition and restricted it only the ones who present adjustment because of some features related to their personality, family functioning or even the prognosis of their child condition. Did the authors explore the profile of the potential participants who refused to participate?
Authors should take these limitations in consideration and go further in its discussion.
Some other minor details must be addressed. In the participants section, it would be important to add more information, namely regarding the cancer diagnosis of the child, and some features of the caregivers. The authors have that information on point 3.1 characteristics of the caregivers, p. 6, they may consider to reallocate this information.
In regression analysis, the variables are predictors or independent, not dependent (line 209)
Reviewer 3 Report
I think something more can be said about the resilience differences associated with psychosocial variables, as religion and marital status.
Some sentences in different sections of the text are too long.
Please, note: there is a redundancy in the keywords; pg 4, line 157 and line 170, the Table 2 references are wrong; also table 2 references on lines 237 & 238 are wrong.
Best regards
Reviewer 4 Report
An article of great interest for improving the health families caregivers of children with cancer. I present some suggestions for improvement:
a) in line 176 to 183, this text should go to point 2.4 Ethical considerations.Otherwise there is a repetition of the ethical criteria of the study;
b) in line 376, I think it would be important, in the suggestion of future studies, to indicate the question how the health system, and health professionals, can contribute to help these families. Is there a need to deepen studies to see how families are resilient with good health systems and health professionals attentive to this aspect?;
c) In conclusion, I think it would be important for the authors to suggest, within the framework of their social contexts, who should promote the "programs focused on promoting the physical and emotional health of their families in this context..."(line 387).
Round 2
Reviewer 1 Report
Thank-you for sharing your revised manuscript. Although you have attempted to address my concerns, the addition to the manuscript is cursory and the revised manuscript still lacks the depth & critical consideration required to make a meaningful contribution to the literature. Consequently I cannot recommend publication.